# Recent Advances in the Aging Microenvironment of Breast Cancer

**DOI:** 10.3390/cancers14204990

**Published:** 2022-10-12

**Authors:** Xiaocong Jiang, Haixing Shen, Xi Shang, Jianwen Fang, Yuexin Lu, Yue Lu, Jingyan Zheng, Peifen Fu

**Affiliations:** 1Department of Breast Surgery, The First Affiliated Hospital, School of Medicine, Zhejiang University, Hangzhou 310003, China; 2School of Medicine, Zhejiang University, Hangzhou 310003, China; 3Department of Breast and Thyroid Surgery, Cixi People’s Hospital, Cixi 315300, China; 4Department of Breast and Thyroid Surgery, Taizhou Hospital, Zhejiang University, Taizhou 318000, China; 5Department of Breast and Thyroid Surgery, First Affiliated Hospital of Huzhou University, Huzhou 313000, China; 6Department of Breast and Thyroid Surgery, Lishui People’s Hospital, The Six Affiliated Hospital of Wenzhou Medical University, Lishui 323000, China

**Keywords:** breast cancer, aging, tumor microenvironment

## Abstract

**Simple Summary:**

The incidence of breast cancer has increased rapidly in recent years. Aging is one of the risk factors for advanced breast cancer. More and more studies have been conducted on the influence of the aging microenvironment on breast cancer. In this review, we summarize the effects of physical changes in the aging microenvironment, senescence-associated secretory phenotypes, and senescent stromal cells on the initiation and progression of breast cancer and the underlying mechanisms. In addition, we also discuss potential targets for senotherapeutics and senescence-inducing agents in the aging microenvironment of breast cancer. We hope this review can provide some directions for future research on the aging microenvironment in breast cancer.

**Abstract:**

Aging is one of the risk factors for advanced breast cancer. With the increasing trend toward population aging, it is important to study the effects of aging on breast cancer in depth. Cellular senescence and changes in the aging microenvironment in vivo are the basis for body aging and death. In this review, we focus on the influence of the aging microenvironment on breast cancer. Increased breast extracellular matrix stiffness in the aging breast extracellular matrix can promote the invasion of breast cancer cells. The role of senescence-associated secretory phenotypes (SASPs) such as interleukin-6 (IL-6), IL-8, and matrix metalloproteases (MMPs), in breast cancer cell proliferation, invasion, and metastasis is worthy of exploration. Furthermore, the impact of senescent fibroblasts, adipocytes, and endothelial cells on the mammary matrix is discussed in detail. We also list potential targets for senotherapeutics and senescence-inducing agents in the aging microenvironment of breast cancer. In conclusion, this review offers an overview of the influence of the aging microenvironment on breast cancer initiation and progression, with the aim of providing some directions for future research on the aging microenvironment in breast cancer.

## 1. Introduction

With the aging and growth of the population, the incidence and mortality of cancer are quickly increasing worldwide [1,2]. Cancer patients aged ≥ 80 years accounted for 13% of all cancer cases in 2018. By 2050, the figure is expected to reach 20.5% [3]. In recent years, the incidence of breast cancer has increased, and the incidence of breast cancer surpassed that of lung cancer as the most commonly diagnosed cancer in the world in 2020 [2]. Breast cancer is estimated to account for nearly one-third of all new cancer cases among women in the United States in 2022. The lifetime probability of developing invasive breast cancer in American women from 2016 to 2018 was as high as 12.9% [4]. The incidence of advanced breast cancer increases with age [5]. In recent decades, oncological research has focused more on tumor cells themselves. However, these cells are surrounded by a stromal microenvironment that has been shown to significantly affect cancer initiation and progression [6]. Somatic mutations accumulated with age cannot fully explain why age is one of the risk factors for breast cancer. LaBarge et al. proposed that it is necessary to understand the changes in the microenvironment and the epigenetic states during aging [7]. Investigating the impact of aging on the tumor microenvironment is interesting and valuable. In recent years, an increasing number of studies have been conducted on the influence of the aging microenvironment on various cancers, including breast cancer [8,9,10]. Aging is a physiological degradation process characterized by genomic instability, telomere attrition, epigenetic alterations, loss of proteostasis, deregulated nutrient sensing, mitochondrial dysfunction, cellular senescence, altered intercellular communication, and stem cell exhaustion [11]. Aging can be both facilitative and suppressive to tumors. On the one hand, cellular senescence can prevent the proliferation of incipient cancer cells and lower the risk of cancer in young organisms. On the other hand, senescent cells also secrete molecules that can stimulate precancerous cell proliferation [6,12,13]. The tumor microenvironment includes the extracellular matrix (ECM), soluble products, and noncancerous host cells such as immune cells, fibroblasts, adipocytes, and endothelial cells [14,15]. This review summarizes the impact of physical changes in the aging breast extracellular matrix, senescence-associated secretory phenotypes (SASPs), and senescent stromal cells on the initiation and progression of breast cancer. The role of immunocytes in the aging microenvironment is also briefly discussed. Moreover, we list some potential therapeutic targets based on the aging microenvironment of breast cancer.

## 2. The Effect of Aging Microenvironment on Breast Tumorigenesis and Cancer Progression

### 2.1. Aging Breast Extracellular Matrix

The extracellular matrix is a network of proteins and polysaccharides secreted by cells. The main components of the extracellular matrix include collagen, fibronectin, laminins, elastin, proteoglycans, and ECM remodeling enzymes [16]. The most abundant matrix protein in the extracellular matrix is collagen (Figure 1a), which regulates and stabilizes the structural properties of the extracellular matrix [17]. Aging results in thinner and curvier collagen fibers, forming a denser collagen mesh (Figure 1b), resulting in increased breast extracellular matrix stiffness [18]. A study undertaking a biophysical and biochemical assessment of stromal-epithelial interactions showed that stiffening is associated with breast cancer invasion and aggression. For example, basal-like and Her2 tumor subtypes have stiffer ECM than Luminal A and B subtypes [19]. To some extent, the aging breast extracellular matrix causes cancer cells to become more invasive and aggressive.

### 2.2. E-Cadherin

Normal mammary epithelial cells (KTB21) were stained with the stroma to detect E-cadherin (E-CAD), and it was interesting to find that E-CAD was localized to the cell membrane in the young matrix (Figure 1a), while little membrane localization was observed in the aged matrix (Figure 1b) [18]. E-cadherin, which is derived from epithelial tissues, is a membrane protein that mediates cell-cell adhesion and epithelial tissue homeostasis [20]. MDA-MB-231 breast cancer cells in an aged matrix showed greater motility and invasiveness than those in a young matrix [18]. E-cadherin is a known tumor suppressor, but Padmanaban et al. found that although the loss of E-cadherin increases the invasion of cancer cells, it also reduces the proliferation and survival of cancer cells, thus limiting the distant metastasis of cancer cells. They demonstrated that due to the loss of E-CAD, the activity of the tumor growth factor beta (TGF-β) signaling pathway in locally disseminated cancer cells is enhanced, thus increasing ROS levels, promoting apoptosis, and limiting the formation of metastases [21].

### 2.3. Senescence-Associated Secretory Phenotypes

A specific set of pro-inflammatory cytokines, chemokines, growth factors, and proteases secreted by senescent cells is known as SASPs [22]. Doxorubicin at 100 nM can induce senescence in MDA-MB-231 breast cancer cells, and the SASPs of the senescent breast cancer cells include interleukin-6 (IL-6), IL-8, matrix metalloprotease-1 (MMP-1), IL-1α, and granulocyte-macrophage colony-stimulating factor (GM-CSF) [23].

SASPs from the senescent breast cancer cell line MDA-MB-231 and senescent normal epithelial cell line MCF-10A, especially those which highly secrete IL-6 and IL-8, can stimulate the invasion and migration of breast cancer cells in vitro [23]. An in vitro study showed that IL-6 and IL-8 in senescence-conditioned medium (SCM) from senescent foreskin fibroblasts, HCA2, can increase the expression of vimentin, ZEB-1, SNAIL-1, and SNAIL-2/Slug in MCF-7 breast cancer cells. These changes suggest that MCF-7 cells undergo epithelial-mesenchymal transition (EMT), enabling cancer cells with more aggressive tumorigenic potential [24]. Furthermore, IL-6 and IL-8 can induce MCF-7 cell proliferation, and interestingly, it is the specific IL-6/IL-8 ratio that induces proliferation rather than an increase in cytokines [25].

Hwang et al. demonstrated that C-X-C motif chemokine ligand 11 (CXCL11) secreted by senescent human umbilical vein endothelial cells (HUVEC) binds to CXCR3 in MDA-MB-231 cells and promotes the migration and invasion of cancer cells in vitro through the ERK pathway. They also proved that CXCL11 could promote cell proliferation and migration in the other three breast cancer cell lines, MCF-7, MDA-MB-453, and HCC70, through the regulation of AKT or ERK activity [26].

MMPs mainly act as proteolytic enzymes that degrade the extracellular matrix, destroy the histological barrier, and promote the invasion and metastasis of tumor cells. MMP-1 is synthesized by cells, such as macrophages and fibroblasts, and can degrade all types of collagen in the mammary gland [27]. MMP-1 expression was found to be increased in the tumor microenvironment in an aging model of breast cancer. A study of two variants of the MDA-MB-231 human breast cancer cell line (231-BR and 231-BR3) xenografted into the mammary fat pad and blood of nude mice demonstrated that MMP-1 facilitates local growth of breast cancer cells and the formation of brain metastases [28]. The amount of MMP-7 anchored to the surface of aging human mammary epithelial cells (HMEC) was significantly reduced compared to young human mammary epithelial cells, thus decreasing the formation of the active soluble form sHB-EGF. Reduced sHB-EGF is unable to sufficiently activate the ErbB4 receptor, thereby blocking an abnormal HMEC proliferation signal [29]. Moreover, collagenase-3 (MMP-13) can act as an important mediator of tumor cell metastasis. Neuregulins (NRGs), which are ErbB ligands, enhance the expression of MMP-13 through a pathway controlled by ERK1/2 kinases, promoting the distant dissemination of breast cancer cells and the local diffusion of the primary tumor [30]. In the future, more comprehensive studies on the influence of SASP components on breast cancer and the interactions between different SASP components will be required.

### 2.4. Senescent Stromal Cells in the Tumor Microenvironment

#### 2.4.1. Fibroblasts

Fibroblasts play a critical role in tissue homeostasis, cancer progression, inflammation, and fibrosis. Fibroblasts synthesize most of the ECM and secrete paracrine growth factors that affect the growth of cancer cells and normal epithelial cells [31]. Cancer-associated fibroblasts (CAFs), a major component of the tumor microenvironment, are permanently activated fibroblasts implicated in therapy resistance and tumor modulation [32,33]. Lysine demethylase 2A (KDM2A) is highly expressed in CAFs and is involved in the carcinogenesis of breast, ovarian, lung, and gastric cancers [34,35,36,37]. Cytokines such as IL-6 and tumor necrosis factor-alpha (TNF-α) released by breast cancer cells can stimulate the expression of KDM2A in fibroblasts. Increased KDM2A induces senescence of normal mammary fibroblasts and upregulates programmed death-ligand 1 (PD-L1) expression in a p53-dependent manner, both in vitro and in vivo, promoting mammary tumor growth. Senescent mammary fibroblasts stimulate the proliferation of breast cancer cells by secreting IL-6, IL-8, and C-X-C motif chemokine ligand 1 (CXCL1) (Figure 2). Furthermore, knockdown of KDM2A abolished the promotion of breast tumor growth by CAFs in vivo, reduced PD-L1 expression in the stroma, and inhibited NOTCH signaling, which can interfere with tumor angiogenesis [38]. This suggests that inhibition of KDM2A expression is a potential therapeutic target in breast cancer. However, some studies have indicated a tumor suppressor effect of KDM2A in breast cancer. A study suggested that KDM2A inhibits breast cancer cell migration and invasion in vitro, possibly by regulating the transcriptional activity of E2F1 [39]. Another in vitro study showed that gallic acid inhibited rRNA transcription and the proliferation of MCF-7 cells by activating KDM2A. The authors also reported that the effect of KDM2A on cell proliferation depends on environmental and intracellular conditions [40]. Therefore, creating novel therapeutic agents for breast cancer by targeting KDM2A requires more in-depth research. Therapy-induced senescence (TIS) also occurs in fibroblasts. A certain dose of doxorubicin can induce senescence in mouse embryonic fibroblasts (MEFs). Senescent MEFs can promote the proliferation of mouse E0771 breast cancer cells and inhibit the apoptosis of cancer cells in vitro by activating the Akt and ERK signaling pathways [41]. An in vitro study using ionizing radiation to induce premature senescence in human breast stromal fibroblasts found that these prematurely senescent fibroblasts have a down-regulated expression of the small leucine-rich proteoglycan decorin, increasing adverse prognostic factors for breast cancer [42]. The fibroblasts used in this study were derived from human breast stroma, which can more accurately reflect the changes in fibroblast behavior in the breast stroma of patients with breast cancer during radiotherapy. However, more comprehensive studies on fibroblast changes in the breast stroma during radiotherapy are needed. Different cell types respond differently to senescence-inducing agents. Under the same conditions, mammary luminal cells are more sensitive to premature senescence induced by H2O2 or gamma rays than fibroblasts in the adjacent stroma. Senescent luminal cells (SLCs) activate mammary stromal fibroblasts primarily via the JAk2/STAT3 pathway in a paracrine manner mediated by IL-8. Activated fibroblasts can enhance EMT and stemness in breast cancer cells, both in vivo and in vitro [43].

#### 2.4.2. Adipocytes

Adipocytes are the most abundant components of breast tissue. Adipocytes have important functions in the breast stroma, such as storing energy, maintaining metabolic homeostasis, secreting adipokines, and providing physical support to mammary epithelial cells [44]. In vitro studies have shown that mature adipocytes can induce EMT in breast cancer cells, promoting the migration, invasion, and proliferation of cancer cells [45,46,47]. TAZ, a key effector of the Hippo signaling pathway, induces resistin expression and secretion in mature adipocytes, thereby promoting breast cancer cell proliferation and maintaining stemness [48]. Moreover, extracellular vesicles (EVs) released from adipocytes can functionally increase the metastatic potential of breast cancer cells in mice via hypoxia-inducible factor-1α (HIF-1α) [49]. An in vitro study also found that EVs from the adipose tissue of obese patients can promote the migration and invasion of MDA-MB-231 cells through the PI3K/AKT pathway [50]. These studies indicate that mature adipocytes in the breast stroma are closely related to breast cancer progression and make for potential therapeutic targets. In postmenopausal women, there is a decrease in the parenchymal composition of the breast, an increased proportion of adipose tissue, and larger adipocytes than in premenopausal women [51]. Senescent adipocytes may partly explain why hormone-dependent subtypes of breast cancer occur more frequently with age as circulating estrogen levels decline. Hypertrophic adipocytes in the breast fat pad of older women cause the recruitment of immunocytes, leading to white adipose tissue (WAT) inflammation. WAT inflammation increases the local production of estrogen, leading to an increased incidence of estrogen-dependent breast cancer in postmenopausal women [51]. A case-control study suggested that adipocyte-derived soluble tumor necrosis factor receptor 2 (sTNF-R2), an inflammatory marker, is an important factor in the increased risk of obesity-induced postmenopausal breast cancer [52]. The protein deleted in breast cancer-1 (DBC1) is a coactivator of nuclear receptors and is involved in the negative regulation of epigenetic modifiers [53]. DBC1 is involved in cellular senescence during obesity by inhibiting histone deacetylase 3 (HDAC3), driving inflammatory responses. At the same time, DBC1 knockout controls senescence and inflammation in the preadipocytes of obese mice [54]. The modulation of HDAC3 activity may be a potential anti-senescence strategy.

#### 2.4.3. Endothelial Cells

Distant metastasis is an important cause of death in patients with breast cancer, and hematogenous metastasis can occur early in breast cancer. In primary breast tumors, the hematogenous dissemination is mediated by three-cell complexes (known as the tumor microenvironment of metastasis), including a perivascular macrophage, a tumor cell overexpressing the actin-regulatory protein Mammalian enabled (Mena), and an endothelial cell (EC). Distant dissemination of lymph node metastases begins through the hematogenous route [55]. An aged endothelial cell monolayer can enhance tumor cell penetration compared to a young endothelial cell monolayer. Price et al. investigated the effect of senescent endothelial cells on breast cancer metastasis using rat brain microvascular endothelial cells (BMECs) isolated from young rats (1-month-old) and aged rats (approximately 24-months-old), which can reflect the characteristics of endothelial cells during natural aging. They found that vascular endothelial growth factor (VEGF) significantly increased the permeability of the aged endothelial monolayer [56]. Sunitinib, an anti-angiogenic drug, has been approved for some tumors, such as advanced renal cell carcinoma. However, many patients are inherently resistant to sunitinib, such as those with metastatic breast cancer (MBC) [57]. Sunitinib does not directly damage ECs but induces resistance and accelerates the metastasis of breast cancer by inducing the phenotype of senescent ECs. Senescent ECs increased the secretion of inflammatory chemokines such as CCL6, chemerin, and IL16. In addition, they can also stimulate the expression of the vascular cell adhesion molecule-1 (VCAM1), which leads to the attraction of MBC cells to ECs and promotes the interactions between tumor cells and ECs, which increases the risk of cancer metastasis. In addition, aging decreases the expression of vascular endothelial cadherin (VEC), thereby undermining junctions between ECs and increasing tumor cell extravasation [58]. The NOTCH signaling pathway plays an important role in angiogenesis, cell proliferation, and differentiation, and NOTCH proteins are highly expressed in many cancers [59]. Notch1 protein (N1ICD) expression is higher in breast cancer tissues than in healthy control tissues. N1ICD of endothelial cells in primary tumors can lead to a senescence-like, pro-inflammatory EC phenotype, promoting tumor cell intravasation and extravasation at distant sites, such as lung metastasis [60]. Blocking the activation of the NOTCH signaling pathway may be an effective target for inhibiting hematogenous metastasis in breast cancer.

#### 2.4.4. Immunocytes in the Aging Microenvironment

Immunocytes in breast tissue are not only vital in immune surveillance but also have a non-negligible impact on normal breast tissue growth and development. At different stages of mammary gland development, immunocytes (such as macrophages, neutrophils, mast cells, T lymphocytes, and B lymphocytes) perform different functions [61]. For example, macrophages can promote the proliferation and differentiation of epithelial cells by producing growth factors and chemokines [62]. Immunocytes within the tumor microenvironment can be divided into two types: tumor-antagonizing immunocytes and tumor-promoting immunocytes [63]. For instance, CD8+ cytotoxic T cells, CD4+ T helper cells, natural killer cells, and M1 macrophages have tumor-antagonizing effects, whereas Treg cells and M2 macrophages are tumor-promoting cells [64]. The immune system also degenerates after a certain age. In this context, immunosenescence is the process of immune system dysfunction during aging. Immunosenescence mainly manifests as persistent low-grade inflammatory responses, impaired responsiveness to neoantigens, and an increased incidence of autoimmune responses [65]. In addition to the natural senescence of immunocytes, other factors can also accelerate their senescence, such as cancer cells and low doses of ionizing radiation [66]. Because macrophages and T lymphocytes are abundant around the ducts and alveoli of normal breast epithelia, the importance of immune-epithelial cell interactions has been suggested [67]. Therefore, here, we focus on macrophages and T lymphocytes in the aging microenvironment.

Macrophages are generally considered the most abundant immunocytes in solid tumors. Macrophages are highly plastic and can differentiate into two “activation” states—pro-inflammatory (M1) macrophages and anti-inflammatory (M2) macrophages [68]. A study based on elderly breast cancer patients found that, compared to the younger control group, breast cancer patients aged older than 80 had a poorer tumor immune microenvironment and significantly higher infiltration of M2 macrophages [10]. M2 macrophages can promote the metastasis of breast cancer cells in vitro and in vivo by secreting chitinase 3-like protein 1 (CHI3L1), which interacts with interleukin-13 receptor α2 chain (IL-13Rα2) molecules on the plasma membrane of cancer cells [69]. Furthermore, chemotaxis and phagocytosis of macrophages decrease with age [70]. The conventional view holds that more abundant immune cell infiltration can inhibit cancer progression. However, a study based on four triple-negative and luminal B breast cancer models found that young mice with more microglia had a greater risk of breast cancer with brain metastases than older mice [71]. Further experiments showed that Sema3a secreted by microglia could stimulate the proliferation and movement of cancer cells, which may be one of the potential reasons for this phenomenon [71].

T lymphocytes, particularly cytotoxic T cells, mainly exert tumor-antagonizing activities. However, senescent T lymphocytes in the body adversely affect the immune system. In breast cancer, the hypoxic microenvironment created by tumor cells allows them to produce more endogenous metabolic cAMP molecules, which can induce T-lymphocyte senescence [72]. This is one of the strategies used by malignant tumors to evade immune responses. However, in vivo and in vitro experiments have shown that activation of toll-like receptor 8 (TLR8) signaling can block this mechanism to suppress T lymphocyte senescence and enhance antitumor immunity in the microenvironment, providing a new strategy for tumor immunotherapy [72,73]. Immunoglobulin-like transcript 4 (ILT4) is an immunosuppressive receptor mainly expressed in myeloid cells, such as dendritic cells, neutrophils, and macrophages. Studies have found that ILT4 is highly expressed in various tumor cells and is associated with immunosuppression [74,75,76]. ILT4 can promote lipid metabolism in tumor cells by activating ERK1/2 signaling and inducing T lymphocyte senescence. Interestingly, in vivo experiments using breast cancer tumor models have shown that blocking ILT4 can significantly prevent tumor-specific T lymphocyte senescence and enhance the effects of tumor immunotherapy [74].

## 3. Treatment Strategies Based on the Aging Microenvironment

### 3.1. Lysyl Oxidase (LOX) Inhibitors

LOX enhances the stiffness and stability of the extracellular matrix by catalyzing the covalent cross-linking of collagen and elastin [77]. The lysyl oxidase family consists of five proteins: LOX and lysyl oxidase like-1 through 4 (LOXL1-LOXL4) [78]. Aging ECM can promote both KTB21 and MDA-MB-231 cell motility and invasion by upregulating the expression of LOX and LOXL2 [18]. LOX active protein and E-CAD protein production were reduced after LOX knockdown with LOX siRNA, resulting in a noticeable reduction in the migration of KTB21 and MDA-MB-231 cells in the aging matrix [18]. LOX enzymes are effective targets for breast cancer treatment based on the aging microenvironment. Moreover, an increasing number of studies have demonstrated the function of LOX inhibitors in the treatment of breast cancer. In vivo experiments have shown that the LOX inhibitor BAPN overcomes chemotherapy resistance in highly aggressive triple-negative breast cancer, especially by inhibiting the HIF-1α/miR-142-3p/LOX/ITGA5/FN1 axis [79]. This result provides hope for triple-negative breast cancer patients with advanced chemotherapy resistance.

### 3.2. Senotherapeutics

Senotherapeutics are composed of senolytics that clear senescent cells and senostatics that neutralize the detrimental effects of the SASP [80]. Currently, these drugs can be divided into synthetic anti-senescence drugs and natural compounds that exist in nature. Information on the safety and efficacy of these drugs is incomplete, and their use as anti-senescence drugs still requires extensive human clinical trials [81,82].

#### 3.2.1. Senolytics

GL-V9 (5-hydroxy-8-methoxy-2-phenyl-7-(4-(pyrrolidin-1-yl) butoxy)-4-H-chromen-4-one), a new synthetic flavonoid derived from wogonin, can efficiently induce apoptosis of senescent MDA-MB-231 and MCF cells in vitro through a ROS-dependent mechanism. More importantly, the inhibitory effect of GL-V9 on senescent cancer cells is similar to that of the classic anti-senescence combination, quercetin + dasatinib, and better than that of quercetin alone [83]. Hence, GL-V9 is a promising antisenescence agent.

TNF-related apoptosis-inducing ligand (TRAIL) is a cytokine secreted by a variety of normal cells that induces exogenous apoptosis in multiple cancer cells through death receptors 4 and 5 (DR4/TRAIL-R1 and DR5/TRAIL-R2). Strong upregulation of DR5 was observed on the surface of prematurely senescent MDA-MB-231 cancer cells treated with doxorubicin or ionizing radiation. In vitro experiments showed that the DR5-selective variant TRAIL D269H/E195R (DHER) significantly enhanced caspase-dependent apoptosis in senescent MDA-MB-231 cancer cells without apparent toxicity to normal cells [84]. This suggests that targeting the DR5 receptor may be a novel cancer treatment strategy with low toxicity.

A recent study found that senescent cells can evade immune surveillance by reducing the expression of NKG2D (one of the NK cell-activating receptors) on the surface of natural killer (NK) cells and cooperating with MMPs to shed NKG2D ligands [85]. This indicates that therapies that block the shedding mechanism of NKG2D ligand may effectively address the risk of senescent cell accumulation in the mammary stroma.

Research on the potential targets of these anti-senescence strategies is not yet mature, but these targets also provide valuable directions for future research. The successful development of senolytics not only provides new targeted treatment options but can also act synergistically with chemotherapy drugs to improve the efficacy of breast cancer treatment regimens.

#### 3.2.2. Senostatics

Apigenin, with its antioxidant and antitumor effects, is a dietary flavonoid widely found in fruits and vegetables, such as parsley, oranges, and tea [86]. The conditioned media (CM) containing SASP produced by ionizing radiation-induced senescence of primary human primary fibroblasts (HCA2 from neonatal foreskin) stimulated the proliferation of relatively aggressive MDA-MB-231 and non-aggressive ZR75.1 human breast cancer cells in vitro. Apigenin markedly reduced the ability of SASP to stimulate the proliferation of breast cancer cells in vitro. In addition, apigenin can help senescent ZR75.1 cells induce a higher expression of tight junction protein ZO-1 and epithelial cytoskeleton protein keratin(K)-18 and lower vimentin expression, thereby indirectly inhibiting the aggressive phenotype of SASP-stimulated breast cancer cells [87].

The murine double minute 2 (MDM2) oncogene is an important negative regulator of the p53 tumor suppressor. By binding to its N-terminal [88], MDM2 can adjust the stability and activity of p53. MI-63 is a small-molecule MDM2 inhibitor that can reactivate p53 in cancer cells and inhibit the activity of NF-κB by effectively blocking MDM2–p53 protein-protein interactions in cells [89]. A study has shown that MI-63 is more effective than rapamycin in decreasing the secretion of IL-6 (a SASP factor that can promote malignant phenotypes) in the mammary epithelial cells MCF10A and 184A1a. SASP can induce EMT in the non-aggressive breast cancer cell line ZR75-1 to promote cancer cell aggressiveness, which can also be inhibited by MI-63 [90]. In other words, MI-63 may partially inhibit the progression of breast cancer by reducing SASP production. However, the pharmacokinetic properties of MI-63 are not ideal, and its bioavailability is not high. Furthermore, MDM2 is not the only protein involved in the regulation of p53, and drug resistance after treatment with MDM2 inhibitors should also be considered.

The mammalian target of rapamycin (mTOR) signaling pathway plays a critical role in mammalian aging. In specific, mTOR is a serine/threonine-protein kinase that belongs to the PI3K-related kinase (PIKK) family, which includes two mTOR complexes, mTOR complex 1 (mTORC1) and 2 (mTORC2) [89]. By inhibiting autophagy and increasing the deposition of damaged proteins and organelles in the body, mTORC1 induces cellular senescence and related diseases [91]. A well-known study showed that inhibiting mTOR exerts anti-senescence and anticancer effects by specifically downregulating the translation of MAPKAPK2 to inhibit the expression of SASP [92]. However, mTOR inhibition also impairs SASP-related immune surveillance, which can be detrimental in the early stages of cancer. The three currently FDA-approved rapamycin analogs (sirolimus, everolimus, and temsirolimus) cause side effects, such as immunosuppression and metabolic disruption, limiting their use in anti-senescence treatments. Lamming et al. screened compound DL001 from approximately 90 rapamycin analogs, which has shown highly selective in vivo inhibition of mTORC1 activity and low inhibition of mTORC2, significantly reducing the occurrence of side effects [93]. No further findings on DL001 have been published yet, and it would be interesting and meaningful to investigate the potential of DL001 in breast cancer treatment.

### 3.3. Senescence-Inducing Agents

Classic cancer treatment regimens exert their effects by targeting rapidly dividing cancer cells with large doses of drugs or radiation, but they may also cause cellular senescence in the surrounding normal cells and cancer cells, known as therapy-induced senescence (TIS). Inducing cellular senescence not only causes growth arrest in cancer cells but also activates antitumor immune responses. Therefore, senescence-inducing agents are expected to be powerful anticancer therapies. Wogonin, an extract from Scutellaria baicalensis, has anti-inflammatory, antiviral, anticancer, and neuroprotective properties [94]. In nude mice with subcutaneously xenografted tumors, moderate concentrations of wogonin can induce senescence of MDA-MB-231 and 4T1 cells without causing severe side effects in normal tissues such as the liver and lung. More importantly, wogonin-induced senescent MDA-MB-231 cells promote macrophage M1 polarization and recruit more M1-like macrophages, thereby targeting and eliminating cancer cells [95]. However, when senescence-inducing agents are reduced or eliminated, cancer cells can escape growth arrest and acquire novel stem cell-related properties that make them more aggressive [96]. The strategy of combining senescence-inducing agents and senolytics could reduce the risks posed by senescence-inducing agents alone. For example, an in vivo study used palbociclib to induce senescence in 4T1 murine breast cancer cells and navitoclax (a Bcl-2 family inhibitor) encapsulated by oligosaccharide-capped nanoparticles to deplete senescent cancer cells. Indeed, the researchers observed enhanced antitumor efficacy, reduced metastasis, and reduced navitoclax off-target toxicity [97].

## 4. Discussion

Human aging is not only reflected in the macroscopic changes of various organs and systems but also in the accumulation of senescent cells in the body and changes in the microenvironment. The role of the aging microenvironment in the initiation and progression of cancer is increasingly valued. The effects of the aging microenvironment on cancer can be attributed mainly to physical changes in the stromal microenvironment, SASPs, senescent stromal cells, and alterations in the immune microenvironment.

Physical changes in the aging microenvironment of the breast have important implications for breast cancer progression. Specifically, increased stiffness of the aging breast matrix promotes breast cancer cell invasiveness. In the stromal microenvironment, SASPs, such as IL-6, IL-8, and MMP-1, can stimulate the proliferation and metastasis of breast cancer cells. The interaction between senescent stromal cells and cancer cells in the tumor microenvironment is also of great interest. In this context, breast cancer cell-derived cytokines can stimulate the expression of KDM2A in normal human breast fibroblasts, and KDM2A can induce fibroblast senescence and release more cytokines to promote the proliferation of breast cancer cells, thus forming a vicious cycle [38]. There is not much research on the effects of senescent adipocytes and endothelial cells in the tumor microenvironment on breast cancer. Adipocytes are the most abundant components of breast tissue. In addition, endothelial cells are closely related to hematogenous metastasis. Both of them deserve more research into the aging microenvironment of breast cancer. As different immunocyte subtypes have contributory or inhibitory effects on tumors, there is abundant research on the effects of immunocytes on breast cancer. In this review, we briefly describe the importance of macrophages and T lymphocytes in the aging microenvironment of breast cancer. According to the results of the present studies, senescent stromal cells mainly promote the occurrence and progression of breast cancer. The complex effects of senescent stromal cells on breast cancer require further study. It is worth noting that the change in hypoxia in the bone marrow microenvironment during aging is closely related to the prognosis of breast cancer. Hypoxia can regulate cellular senescence, and HIF-1α can promote tumor progression by affecting the expression of specific gene networks under hypoxia. Therefore, the influence of changes in the bone marrow microenvironment during aging on breast cancer deserves more attention [98]. In addition, we summarize some of the latest therapeutic ideas targeting the aging microenvironment of breast cancer. The strategy of combining senescence-inducing agents and senolyticsis is promising. These therapeutic ideas will provide directions for future research and further deepen research on the aging microenvironment of breast cancer.

The number of studies on the aging microenvironment continues to grow. In the process of writing this review, we found that research on the aging microenvironment of breast cancer has problems that cannot be ignored. Due to the limitations of many factors, such as ethical and economic issues and natural lifespan, most aging models used in breast cancer research are induced by oncogenes, ionizing radiation, or drugs. Therefore, they may be unable to accurately reflect natural human aging, and conclusions drawn from naturally aged human breast tissues are invaluable. No detailed studies have compared the composition and changes in the aging microenvironment of commonly used aging models and human breasts. Such comparative studies will provide important guidance for the selection of aging models in future experiments.

## 5. Conclusions

The effects of the aging microenvironment on breast cancer are complex. However, overall, it promotes the initiation and progression of breast cancer. Many potential therapeutic targets have been proposed based on the aging microenvironment of breast cancer. These therapeutics require further studies on their safety, efficacy, and interactions with existing therapeutics. In conclusion, valuable progress has been made in the study of the aging microenvironment of breast cancer. However, several new issues remain to be explored.

## Figures and Tables

**Figure 1 cancers-14-04990-f001:**
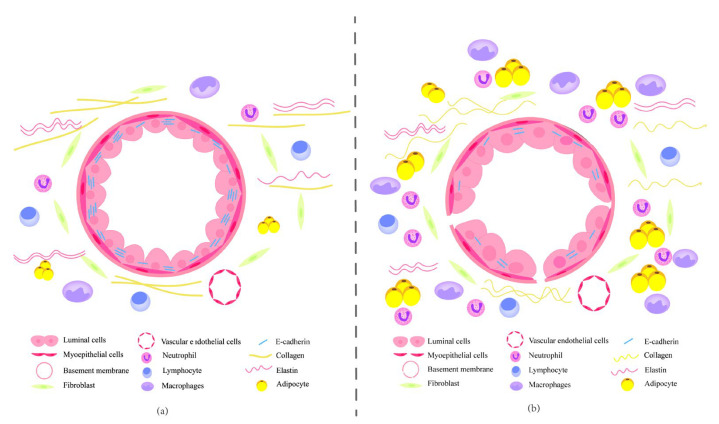
(**a**) Mammary microenvironment (young). The mammary microenvironment contains the extracellular matrix (ECM), immune cells, fibroblasts, adipocytes, and endothelial cells. E-cadherins are highly expressed in the membrane of luminal cells. (**b**) Mammary microenvironment (aged). Aging causes collagen to become thinner and curvier, forming a denser collagen mesh and increasing the stiffness of the extracellular matrix. The expression of E-cadherins decreases in the membrane of luminal cells.

**Figure 2 cancers-14-04990-f002:**
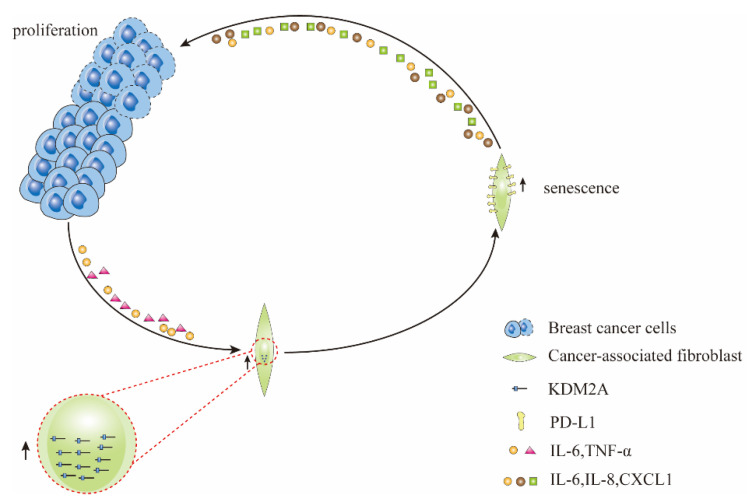
Cytokines, such as interleukin (IL)-6 and tumor necrosis factor-alpha (TNF-α), secreted by breast cancer cells, can stimulate the expression of KDM2A in cancer-associated fibroblasts, which induces the senescence of cancer-associated fibroblasts and upregulates the expression of programmed death-ligand 1 (PD-L1). PD-L1 and cytokines such as IL-6, IL-8, and C-X-C motif chemokine ligand 1 (CXCL1) secreted by senescent cancer-associated fibroblasts can promote the growth and proliferation of tumor cells.

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
