# Peer review of "Recent Advances in the Aging Microenvironment of Breast Cancer"

_cancers, 2022, doi:10.3390/cancers14204990_

Round 1

Reviewer 1 Report

The review article is well organized and highlights the major research developments in the aging microenvironment of breast tissue. I agree that the role of the aging microenvironment in the initiation and progression of cancer is increasingly valued. Especially since research focusing on the mechanobiology of cells have provided a lot of evidence in understanding disease development.

The authors have great discussion points and I believe the overall scope of the review is apt for the journal. It will also be of interest for a wide range of readership. I recommend the article with a couple of minor revisions.

I recommend adding the following references to the review which would add more value,

1.      Ferrer, Alejandra, et al. "Hypoxia-mediated changes in bone marrow microenvironment in breast cancer dormancy." Cancer Letters 488 (2020): 9-17.

2.      LaBarge, Mark A., et al. "Breast cancer beyond the age of mutation." Gerontology 62.4 (2016): 434-442.

3.      Chaturvedi, Sukhada, and Ralf Hass. "Extracellular signals in young and aging breast epithelial cells and possible connections to age-associated breast cancer development." Mechanisms of ageing and development 132.5 (2011): 213-219.

Combine Figures 1 and 2 together for better comparison. Figure 3 needs to be moved to where it is mentioned in the text.

Minor typographical errors need to be corrected. Ex. Line 120, “which highly secret IL-6 and IL-8,” should read secrete. All in vitro and in vivo text should be italicized.

Reviewer 2 Report

In this manuscript, the authors have discussed the effect of tumor microenvironment in aging on breast cancer tumorigenesis and progression. They firstly described the roles of individual component from aging tumor microenvironment, including extracellular matrix, E-cadherin from epithelial cells, senescence-associated secretory phenotypes, and stromal cells (fibroblast, adipocytes, endothelial cells, and immunocytes). Next, they discussed the current and possible aging microenvironment targeted drugs. In addition, they mentioned that aging models should be compared for future studies. Overall, they concluded that treatment on aging microenvironment could be an effective way for breast cancer.  

1) From Fig 1 and 2, do any change on other components other than collagen and e-cadherin? It seems like the only difference is the morphological change on collagen and amount of E-cadherin between young and aging mammary microenvironment. The rest are identical. 

Reviewer 3 Report

The manuscript "Recent advances in the aging microenvironment of breast cancer" is an interesting study on a clinically very important issue, which is the microenvironment of breast cancer cells.                                          Manuscript is developed synthetically, and the references chapter contains the vast majority of well-selected publications from the last 10 years.     Figures 1 and 2 are also very important.                                                          Due to the clinical aspect of the manuscript and the brief study of the issue of the microenvironment of breast cancer cells "Recent advances in the aging microenvironment of breast cancer", I propose to accept it for publication in CANCERS in its current form.

Author Response

We would like to thank you for your careful reading and helpful comments.